# Metabolomic Profiling of Plasma and Erythrocytes in Sickle Mice Points to Altered Nociceptive Pathways

**DOI:** 10.3390/cells9061334

**Published:** 2020-05-26

**Authors:** Klétigui Casimir Dembélé, Thomas Mintz, Charlotte Veyrat-Durebex, Floris Chabrun, Stéphanie Chupin, Lydie Tessier, Gilles Simard, Daniel Henrion, Delphine Mirebeau-Prunier, Juan Manuel Chao de la Barca, Pierre-Louis Tharaux, Pascal Reynier

**Affiliations:** 1Faculté de Pharmacie, Université des Sciences, des Techniques et des Technologies de Bamako BP, Bamako 1805, Mali; kletigui@outlook.fr; 2Département de Biochimie et Génétique, Centre Hospitalier Universitaire, 49933 Angers, France; Floris.Chabrun@chu-angers.fr (F.C.); Stephanie.Chupin@chu-angers.fr (S.C.); LyTessier@chu-angers.fr (L.T.); GiSimard@chu-angers.fr (G.S.); DePrunier@chu-angers.fr (D.M.-P.); JMChaoDeLaBarca@chu-angers.fr (J.M.C.d.l.B.); 3Unité Mixte de Recherche (UMR) MITOVASC, Centre National de la Recherche Scientifique (CNRS) 6015, Institut National de la Santé et de la Recherche Médicale (INSERM) U1083, Université d’Angers, 49933 Angers, France; daniel.henrion@univ-angers.fr; 4Paris Cardiovascular Centre (PARCC), Institut National de la Santé et de la Recherche Médicale (INSERM), 75015 Paris, France; carmintz@gmail.com; 5Institut National de la Santé et de la Recherche Médicale (INSERM) U1253, Université François Rabelais de Tours, 37000 Tours, France; C.VEYRATDUREBEX@chu-tours.fr; 6Laboratoire de Biochimie et Biologie Moléculaire, Centre Hospitalier Universitaire de Tours, 37000 Tours, France; 7Université Paris Descartes, Sorbonne Paris Cité, 75006 Paris, France; 8Nephrology Division, Georges Pompidou European Hospital, Assistance Publique-Hôpitaux de Paris, 75015 Paris, France

**Keywords:** metabolomics, lipidomics, nociception, sickle cell disease

## Abstract

Few data-driven metabolomic approaches have been reported in sickle cell disease (SCD) to date. We performed a metabo-lipidomic study on the plasma and red blood cells of a steady-state mouse model carrying the homozygous human hemoglobin SS, compared with AS and AA genotypes. Among the 188 metabolites analyzed by a targeted quantitative metabolomic approach, 153 and 129 metabolites were accurately measured in the plasma and red blood cells, respectively. Unsupervised PCAs (principal component analyses) gave good spontaneous discrimination between HbSS and controls, and supervised OPLS-DAs (orthogonal partial least squares-discriminant analyses) provided highly discriminant models. These models confirmed the well-known deregulation of nitric oxide synthesis in the HbSS genotype, involving arginine deficiency and increased levels of dimethylarginines, ornithine, and polyamines. Other discriminant metabolites were newly evidenced, such as hexoses, alpha-aminoadipate, serotonin, kynurenine, and amino acids, pointing to a glycolytic shift and to the alteration of metabolites known to be involved in nociceptive pathways. Sharp remodeling of lysophosphatidylcholines, phosphatidylcholines, and sphingomyelins was evidenced in red blood cells. Our metabolomic study provides an overview of the metabolic remodeling induced by the sickle genotype in the plasma and red blood cells, revealing a biological fingerprint of altered nitric oxide, bioenergetics and nociceptive pathways.

## 1. Introduction

Sickle cell disease (OMIM **#**603903) is the most common autosomal recessive disorder worldwide, caused by the homozygous pathogenic variant p.(Val6Glu) in the gene encoding the β-globin chain of hemoglobin (*HBB*) [1]. The increased tendency of hemoglobin S to polymerize and aggregate, enhanced in circumstances favoring hypoxia and hemoglobin deoxidation, increases the erythrocyte rigidity, producing the “sickled” red blood cells (RBC) with altered rheological properties. This leads to hemolysis, anemia and painful life-threatening vaso-occlusive crises with deleterious consequences in the organs, such as pulmonary hypertension, ischemic stroke, cutaneous leg ulceration, priapism, and renal insufficiency.

Metabolomics, the next generation of metabolic biochemistry, is a powerful hypothesis-free approach highlighting the biological imprinting of the diseases. Surprisingly, very few metabolomic studies have been performed in sickle cell disease so far [2,3]. The most documented study, performed by Darghouth et al. [4], compared the RBC from 28 adult patients with the HbSS genotype with those of 24 healthy adults (HbAA), using a chromatography-mass spectrometry based untargeted metabolomic method. Among the 89 metabolites identified, 31 exhibited significantly modified concentrations, revealing an involvement of glycolysis, pentose phosphate, glutathione, ascorbate, amino acids, polyamines, carnitine, and creatine metabolisms in the pathogenesis. Using metabolomic profiling in the whole blood and plasma of a mouse model of the disease, the Xia group progressively deciphered a central pathogenic mechanism, also found in humans, promoting the polymerization of the hemoglobin S [5,6]. They showed that increased concentrations of adenosine resulted, through the activation of the adenosine A2B receptor (ADORA2BR), in elevated concentrations of 2,3-biphosphoglycerate (2,3-DBG) and sphingosine-1-P (S1P). This activation was, however, pharmacologically reversible. The same group later showed that elevated concentrations of S1P mediated multi-tissue chronic inflammation through IL6 [7]. Lastly, a proton nuclear magnetic resonance (^1^H-NMR)-based metabolomic approach was applied in the plasma of patients (*n* = 23) with SCD according to their level of albuminuria [8]. The concentration of six metabolites was modified according to albuminuria: betaine, proline, dimethylarginine, glutamate, leucine, and lysine.

In this study, we used a standardized and quantitative data-driven targeted metabolomics procedure to explore the metabolic fingerprints of the plasma and RBC in a SCD mouse model carrying the human mutated gene (HbSS), compared with wild type (HbAA) and heterozygous (HbAS) genotypes.

## 2. Materials and Methods

### 2.1. Animals and Sampling Procedures

All experiments were performed in accordance with the European Community Guiding Principles for the care and use of animals (Directive 2010/63/UE; Décret n°2013-118). Studies used humanized SCD knock-in mice, with notation B6;129-Hba^tm1(HBA)Tow^Hbb^tm2(HBG1,HBB*)Tow^/Hbb^tm3(HBG1,HBB)Tow/J^, developed by Wu et al. [9]. Experimental animals (HbSS) were homozygous for mutant β globin (E to V at position 6) and expressed human HbS. Control animals (HbAS or HbAA), derived from the same colony, were heterozygous or homozygous for wild-type β globin and expressed human HbA. Our metabolomic analysis was carried out on a total of 56 mice, all studied in steady state, with an age comprised between 9-12-week-old. These mice were divided in two comparative cohorts, one comparing HbSS (*n* = 23; 8 males and 15 females) with HbAS (*n* = 14; 9 males and 5 females) and the other comparing HbSS (*n* = 9; 4 males and 5 females) with HbAA (*n* = 10; 6 males and 4 females) genotypes.

All animals were maintained in a room at constant temperature (21 ± 2 °C) with a 12 h light/dark cycle. They were fasting since the day before blood collection with free access to drinking water. 130 to 150 µL of blood were collected in the morning by the same trained operator from the mandibular vein into a heparinized tube.

Samples were centrifuged for 10 min at 5000× *g* and 4 °C. The supernatant (plasma) was recovered and immediately stored in 50 μL aliquots at -80 °C until analysis. None of the plasma studied showed apparent hemolysis. Then, 50 µL of the RBC pellet were collected and washed with 150 µL of NaCL 0.9%. RBCs cell pellets were immediately stored in 50 μL aliquots at −80 °C until analysis.

### 2.2. Red Blood Cell Extraction

RBC samples (50 μL) were mixed with a cold solution of PBS (15 μL) and methanol (85 µL) and transferred in pre-cooled 2.0 mL homogenization Precellys tubes prefilled with 1.4 mm diameter ceramic beads. Homogenization was performed using a Precellys homogenizer (Bertin Technologies, Montigny-le-Bretonneux, France) kept in a room at +4 °C. The supernatant (cell extract) was recovered after centrifuging the homogenate (10,000× *g*, 5 min at +4 °C) and kept at -80 °C until metabolomic analysis.

### 2.3. Metabolomic Analyses

Targeted quantitative metabolomic analysis was carried out with a Biocrates^®^ Absolute IDQ p180 kit (Biocrates Life Sciences AG, Innsbruck, Austria) and a QTRAP 5500 mass spectrometer (SCIEX, Villebon-sur-Yvette, France) to quantify 188 metabolites. The full list of individual metabolites is available at http://www.biocrates.com/products/research-products/absoluteidq-p180-kit. Sample preparation and analyses were performed following the Kit User Manual.

### 2.4. Statistical Analyses

Metabolites with more than 20% of concentration values below the lower limit of quantitation (LLOQ) or above the upper limit of quantitation (ULOQ) were excluded from the raw metabolomic data before statistical analyses. Each metabolite concentration was normalized by the sum of the concentrations of all the metabolites, similar to normalization by the TIC (total ion current). Multivariate analysis was first performed using unsupervised principal component analysis (PCA) for the detection of sample grouping and outliers. Supervised orthogonal partial least squares-discriminant analysis (OPLS-DA) was then applied to maximize variation between the disease (HbSS) and control (HbAA and HbAS) groups, and to determine the metabolites contributing to this variation. Multivariate analyses for the PCA and OPLS-DA models were performed on mean-centered and unit variance-scaled (MC-UV) data. The quality of the OPLS-DA model was validated by two parameters, namely goodness of fit (R^2^) and goodness of prediction indicated by the cumulated Q^2^ value (Q^2^cum). A threshold of 0.5 was used to determine whether an OPLS-DA model could be estimated as having a good (Q^2^cum ≥ 0.5) or a poor (Q^2^cum < 0.5) predictive capability [10]. VIP (variable importance for the projection) values summarize the importance of each variable for the OPLS-DA model, whereas the loading values are indicators of the relationship between the y vector containing the class information (i.e., HbSS or controls) and variables in the X matrix (i.e., metabolites). In the reduced model, plotting key variables based on VIP values versus loading values scaled as correlation coefficients (*p*_corr_) yields a V-shaped graph known as the “volcano” plot, on which the most important variables are easily recognized. Multivariate data analysis was conducted using SIMCA-P v.14.0 (Umetrics, Umeå, Sweden).

## 3. Results

### 3.1. HbSS/HbAS Plasma Signature

Among the 188 metabolites analyzed by the kit, 153 were accurately measured in the plasma (the 5661 metabolites concentrations are given in Appendix A), including the hexoses (*n* = 1, sum of), amino acids (*n* = 21), biogenic amines (*n* = 15), carnitine, acylcarnitines (*n* = 12), sphingomyelins (*n* = 15), phosphatidylcholines (*n* = 75) and lysophosphatidylcholines (*n* = 13). Unsupervised PCA gave a good spontaneous separation of the two genotypes (R2X = 0.821; Q2cum = 0.637; Figure 1A). Supervised OPLS-DA provided a highly discriminant model for the separation of the two groups (R2X = 0.71, R2Y = 0.945, Q_2_ = 0.797, Figure 1B) with a low risk of over-fitting (CV-ANOVA *p*-value = 7.16 × 10^−8^).

This multivariate model included 29% (*n* = 44) of the accurately measured metabolites with a VIP > 1 (Figure 2). The values of VIPs and loadings are given in Appendix A. These most discriminant metabolites comprised the hexoses (*n* = 1, sum of), nine amino acids (glycine, ornithine, proline, arginine, histidine, methionine, aspartate, alanine, and glutamine), seven biogenic amines (ADMA, total DMA, serotonin, putrescine, SDMA, kynurenine, and spermidine), six acylcarnitines (C3, C14, C16, C16:1, C18, and C18:1), five sphingomyelins, and 16 phosphatidylcholines. The sense of the modification of the concentrations of these metabolites are given in the volcano plot presented in Figure 2.

### 3.2. HbSS/HbAS Red Blood Cells Signature

Among the 188 metabolites analyzed by the kit, 129 were accurately measured in the RBC extracts (the 4773 metabolites concentrations are given in Appendix A), including the hexoses (*n* = 1, sum of), amino acids (*n* = 19), biogenic amines (*n* = 13), carnitine, acylcarnitines (*n* = 8), sphingomyelins (*n* = 14), phosphatidylcholines (*n* = 61), and lysophosphatidylcholines (*n* = 12). Unsupervised PCA gave a good spontaneous separation of the two genotypes (R2X = 0.859; Q_2_ = 0.678; Figure 3A). Supervised OPLS-DA provided a highly discriminant model separation of the two groups (R2X = 0.685; R2Y = 0.898; Q*_2_* = 0.787; Figure 3B) with a low risk of over-fitting (CV-ANOVA *p*-value = 7.42 × 10^−9^).

This multivariate model included 48% (*n* = 62) of the accurately measured metabolites with a VIP > 1 (Figure 4). The values of VIPs and Loadings are given in Appendix A. The most discriminant metabolites comprised the hexoses (*n* = 1, sum of), six amino acids (glycine, glutamine, aspartate, proline, asparagine, and glutamate), four biogenic amines (alpha-AAA, putrescine, taurine, ADMA), carnitine, C3 acylcarnitine, 13 sphingomyelins, 33 phosphatidylcholines, and three lysophosphatidylcholines. The sense of the modification of the concentrations of these metabolites is given in the volcano plot presented in Figure 4.

### 3.3. Summary of the HbSS/HbAS Signature

Whole data visualization is provided in Figure 5 using the word clouds summarizing the plasma and red blood cell signatures of the HbSS/HbAS comparison.

### 3.4. HbSS/HbAA Plasma Signature

Among the 188 metabolites analyzed by the kit, 131 were accurately measured in the plasma (the 2489 metabolites concentrations are given in Appendix A). Unsupervised PCA gave a good spontaneous separation of the two genotypes (R2X = 0.777; Q_2_ = 0.587; Figure 6A). Supervised OPLS-DA provided a highly discriminant model for separating the two groups (R2X = 0.766; R2Y = 0.917; Q_2_ = 0.776; Figure 6B) with a low risk of over-fitting (CV-ANOVA *p*-value = 0.0023).

This multivariate model included 47% (*n* = 62) of the accurately measured metabolites with a VIP > 1 (Figure 7). The value of VIPs and loadings are given in Appendix A. The most discriminant metabolites comprised two amino acids (arginine and threonine), three biogenic amines (ADMA, putrescine and kynurenine), four sphingomyelins, 45 phosphatidylcholines, and eight lysophosphatidylcholines. The sense of the modification of the concentrations of these metabolites is given in the volcano plot presented in Figure 7.

### 3.5. HbSS/HbAA Red Blood Cell Signature

Among the 188 metabolites analyzed by the kit, 113 were accurately measured in the RBC (the 2147 metabolites concentrations are given in Appendix A). Unsupervised PCA gave a good spontaneous separation of the two genotypes (R2X = 0.86; Q_2_ = 0.749; Figure 8A). Supervised OPLS-DA provided a highly discriminant model separation of the two groups (R2X = 0.844; R2Y = 0.908; Q_2_ = 0.728; Figure 8B) with a low risk of over-fitting (CV-ANOVA *p*-value = 0.0066).

This multivariate model included 64% (*n* = 72) of the accurately measured metabolites with a VIP > 1 (Figure 9). The values of VIPs and loadings are given in Appendix A. The most discriminant metabolites comprised 3 amino acids (arginine, histidine, and aspartate), five biogenic amines (ADMA, putrescine, alpha-AAA, spermine, spermidine), carnitine and acylcarnitine C3, 13 sphingomyelins, 44 phosphatidylcholines, and six lysophosphatidylcholines. The sense of the modification of the concentrations of these metabolites is given in the volcano plot presented in Figure 9.

## 4. Discussion

The contribution of nitric oxide (NO) to the vaso-occlusive process in SCD is well established [11,12]. NO is involved in the regulation of vaso-motor tone as well as in platelet aggregation and the adhesion of sickle blood cells to the activated endothelium [12]. Arginine is the precursor of NO, through NO synthases (NOS), but it can also be metabolized to ornithine and urea, through arginase activity, as the ornithine itself is a substrate for polyamines (putrescine, spermine and spermidine), proline and glutamate synthesis. SCD is characterized by a metabolic shift in favor of arginase, to the detriment of NO synthesis, due to the release of arginase from erythrocytes during hemolysis, as well as to inflammatory cytokines which reduce NO availability [12]. Our metabolomic signature is in full agreement with this metabolic shift, showing decreased concentration in the HbSS genotype of arginine in the plasma and red blood cells, whereas the direct or indirect products of arginase, such as ornithine, proline, glutamate, putrescine, spermidine, and spermine, were found to be increased in the plasma. In the HbSS red blood cells, although the putrescine was also increased, spermidine and spermine were diminished. Polyamines are known to be involved in SCD pathophysiology through their association with spectrin, which stabilizes the red blood cell membranes [13].

ADMA (asymetric NG, NG-dimethylarginine), SDMA (symetric dimethylarginine) and total DMA (dimethylarginine) were globally also increased in the HbSS signatures as it has been previously reported in the blood of patients [14]. Both ADMA and SDMA derive from intranuclear methylation of l-arginine residuals and are released into the cytoplasm after proteolysis. Presumably, SDMA is a weak inhibitor of eNOS, most likely weaker than ADMA. SDMA is also a potent competitor of l-arginine transport [15], and it impairs l-arginine uptake from the perfused loop of Henle [16]. This suggests that SDMA may compete with arginine transport and thereby still could have an indirect inhibitory effect of NO synthesis by limiting arginine availability to NOS. In the general population, SDMA, but not ADMA, was associated with cardiovascular mortality and all causes mortality [17]. Circulating ADMA was found to be elevated in several conditions of endothelial dysfunctions [18], as a biomarker of poor prognostic. ADMA being an endogenous NOS inhibitor, in SCD, its increased concentration reinforces the NO deficiency [19]. These results could be due to the balance between increased production through hemolysis and haemoglobin proteolysis as well as to impaired clearance.

NO, arginine, ornithine and ADMA also play a role in pain pathophysiology and the restoration of arginine bioavailability through exogenous supplementation was shown to contribute to the treatment of the disease by increasing the plasma NO concentration and reducing the pain [20,21]. Polyamines are also involved in pain since their increased synthesis related to inflammation was shown to be involved in the development of nociception, a subcutaneous injection of putrescine, spermidine or spermine into the rat paw showing mechanical allodynia and edema [22]. In contrast, a polyamine diet has been shown to relieve pain hypersensitivity in rats [23].

In addition to this metabolomic signature centered on arginine metabolism, we also found that many other amino acids disclosed increased concentrations in the HbSS genotype, mostly in the plasma, such as aspartate, asparagine, alanine, glycine, glutamate, methionine, threonine, glutamine, and taurine. RBC of SCD patients exhibit an abnormally high abundance of N-methyl D-aspartate receptors (NMDAR), mediating excessive calcium uptake which could contribute to the crisis [24]. Interestingly, these authors proposed that glutamate may be found to be increased in this context. We here found that the glutamate, as well as the other NMDAR agonists, aspartate and glycine were all increased in HbSS. In addition to its direct role in sickle erythrocytes, NMDARs, especially those located in the dorsal horn of the spinal cord, are critically involved in nociceptive transmission and central sensitization of pain [25], which could also be impaired in the disease. Taurine, which contributes to the sensitivity to nociceptive chemical stimulation [26], is another amino acid derivative found in our sickle signatures.

Two mediators derived from the essential amino acid tryptophan showed altered concentrations, the kynurenine that was increased in the two plasma signatures and the serotonin that was reduced in the plasma of the HbSS/HbAS signature. These modifications of kynurenine and serotonin have never been reported before in our knowledge in SCD. Kynurenine metabolites such as quinolinic acid, can contribute to the development of depression via NMDA glutamatergic stimulation. The kynurenine is also an immune metabolite involved in inflammation and vasodilatation [27,28]. Interestingly, it was shown that the kynurenine metabolic pathway is an important mediator of neuropathic pain in rats that constitute a therapeutic target [29].

Serotonin or 5-hydroxytriptamine (5HT) is an intermediate product of tryptophan metabolism located primarily in the enterochromaffin cells of the intestines, brain serotonergic neurons, and blood platelets. In tissues such as the brain, inflammation increases tryptophan metabolism, thereby depleting its availability for serotonin production [30]. Serotonin is an important endogenous mediator of the descending anti-nociceptive system in the central nervous system. Indeed, modulation of painful stimuli includes the release of serotonin to inhibit pain transmission. Serotonin modulates the descending fibers from the midbrain to the dorsal horn that inhibit the transmission of the painful stimuli. In mast cells from mouse models expressing human sickle hemoglobin, there was evidence of decreased serotonin released under chronic morphine exposure [31]. Sickle hyperalgia was shown to be modulated by the descending serotonergic anti-nociceptive system in HbSS-BERK (sickle) mice, while the serotonin concentration in the spinal cord dorsal is influenced by diet and correlated with hyperalgia [32]. Serotonin is also known to enhance the interaction of platelets with circulating macrovesicles, to increase platelet activation, and to increase their overall pro-coagulant activity [33]. Taking into account the importance of coagulation and nociception mechanisms in the pathophysiology of the disease, this decreased concentration of serotonin observed in the basal state of the mouse model could either counteract the pro-coagulant state or attest to the nociception deregulation.

The concentration of the sum of hexoses was found to be reduced in the plasma and red blood cells HbSS genotype compared with HbAS. This may reflect a preferential shift to glycolytic metabolism, secondary to the low oxygen levels in tissues due to the reduced oxygen-carrying capacity of HbS. Indeed, in addition, six long-chain acylcarnitines, that are also substrates for mitochondrial oxidation, were also found to be increased in the plasma of HbSS compared with HbAS genotypes. Such accumulation of acylcarnitines has been reported in the two previously reported metabolomic studies performed in RBC in patients [4] and transgenic mice [6]. This glycolytic shift could be associated with slightly increased insulin sensitivity in this mouse model. Indeed, we observed a reduced concentration of alpha-AAA (alpha-aminoadipate) in the two RBC signatures, which was recently identified as a biomarker of insulin resistance [34]. Alpha-AAA is involved in the lysine biosynthetic pathway and has been poorly investigated in pathophysiological conditions. Its reduced concentration, combined with the low levels of the hexoses and increased long chain acylcarnitines, argues in favor of higher insulin sensitivity in this mouse model. Such increased insulin sensitivity has recently been reported in sickle patients [35]. In addition, alpha-aminoadipate is known to reduce nerve injury-induced neuropathic pain in mice through glial signaling [36], indicating its potential additional role in nociception.

Carnitine, known for its antioxidant properties and propionylcarnitine (C3) were increased in the two RBC signatures, as previously reported in RBC from patients [4] and mice [6]. Carnitine and acetyl-carnitine are known to be used as a buffer and reservoir of activated acyl group for the turnover and the repair of RBC membranes during repeated cycled of erythrocyte sickling and unsickling and mechanical fragmentation under shear stress participating in HbS cell membrane fragility [37].

Lastly, the concentrations of many lysophosphatidylcholines, phosphatidylcholines, and sphingomyelins were sharply reduced by the HbSS genotype, both in the plasma and RBC. Reduced concentrations of sphingomyelin and ceramide have already been reported in the serum of patients with SCD [38]. Phosphatidylcholines were also found to be reduced in the erythrocytes from the mouse model [8]. However, in contrast with our results lysophosphatidylcholines were found to be increased in this study. This lipidomic signature could therefore reflect the remodeling of sickle RBC. It could also be involved in pain since an alteration of the phosphatidylcholines, sphingomyelins, and ceramides metabolism was also found by a metabolomic approach in the dorsal horn during chronic neuropathic pain in a rat model [39].

It is interesting to note that this murine model reveals metabolomic features closely related to the signatures observed in patients with sickle cell anemia. We recently reported the metabolomic signatures observed in the plasma and red blood cells of patients during the vaso-occlusive crisis, in comparison with the steady state [40]. Globally, the same groups of metabolites are affected in the patients and mouse model. In the plasma, the signature involving the metabolism of NO and arginine is also in the foreground in patients during the vaso-occlusive crisis, as is the energetic metabolism, the decrease concentration of serotonin and the metabolites associated with pain. In red blood cells, the phospholipid remodeling is important both in humans and mice. However, during the vaso-occlusive crisis in humans, the phosphatidylcholines and lysophosphatidylcholines are mostly increased, while they are drastically reduced in the mouse model in steady state. It is likely that this reflects the sickling of red blood cells occurring during the crisis.” It would be also interesting to explore the sexual dimorphism of these sickle mice metabolomic signatures, which may explain the sex-related specificity of the clinical expression that has been reported [41].

## 5. Conclusions

In conclusion, this quantitative targeted metabo-lipidomic approach performed in plasma and RBC of a mouse model carrying the human HbSS variant revealed new deregulated pathways affecting amino acids, kynurenine, serotonin, alpha-aminoadipate, hexoses, and complex lipids. These metabolic alterations associated with SCD point towards a glycolytic shift and increased insulin sensitivity in this model and to a complex set of metabolites contributing to nociception (NO, arginine, ornithine, ADMA, polyamines, glutamate, aspartate, glycine, taurine, alpha-aminoadipate, kynurenine and serotonin), while the lysophosphatidylcholines, phosphatidylcholines, and sphingomyelins are potentially involved in both pain pathophysiology and RBC remodeling.

## Figures and Tables

**Figure 1 cells-09-01334-f001:**
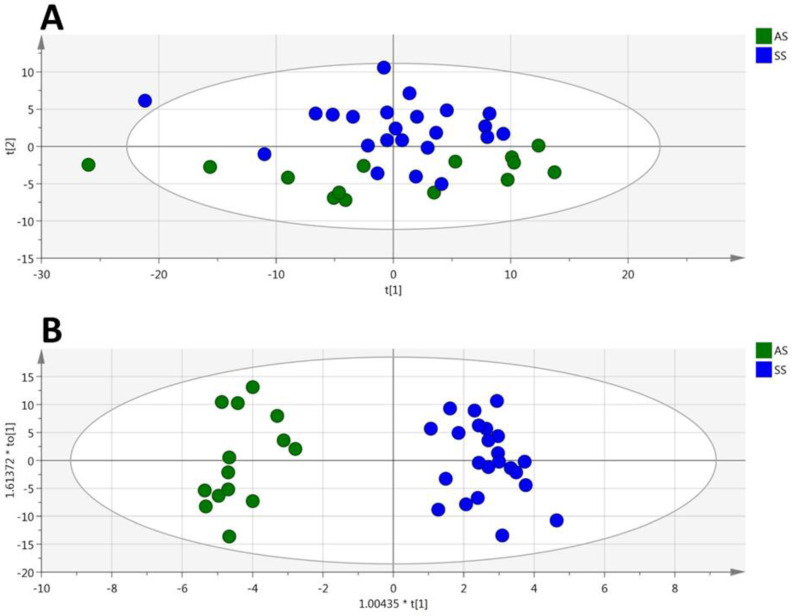
(plasma). PCA (**A**) and OPLS-DA (**B**) scatter plots obtained from the matrix of metabolites measured in the plasma from HbSS (blue circles, *n* = 23) and HbAS (green circles, *n* = 14) mice.

**Figure 2 cells-09-01334-f002:**
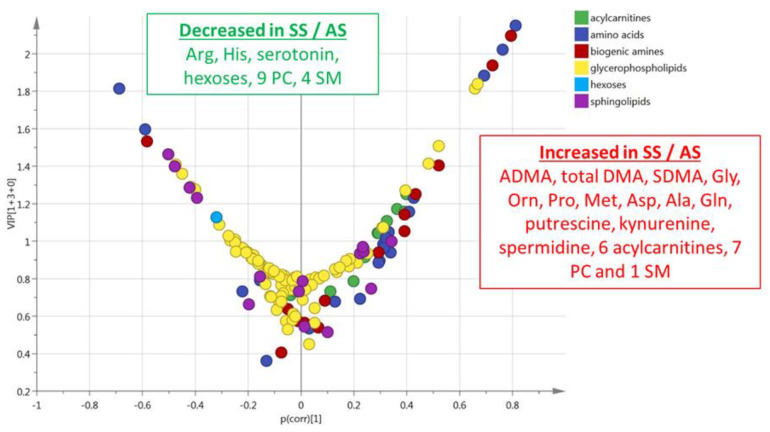
(plasma). Volcano plot (*p*_corr_ vs. VIP) from the OPLS-DA model shown in Figure 1B (plasma from HbSS vs. HbAS). Only the most discriminating metabolites with high VIP values ≥ 1 are listed in the boxes. Negative *p*_corr_ values (left) indicate diminished metabolite concentrations in HbSS versus HbAS genotypes, whereas positive *p*_corr_ values (right) indicate increased metabolite concentrations in HbSS versus HbAS genotypes.

**Figure 3 cells-09-01334-f003:**
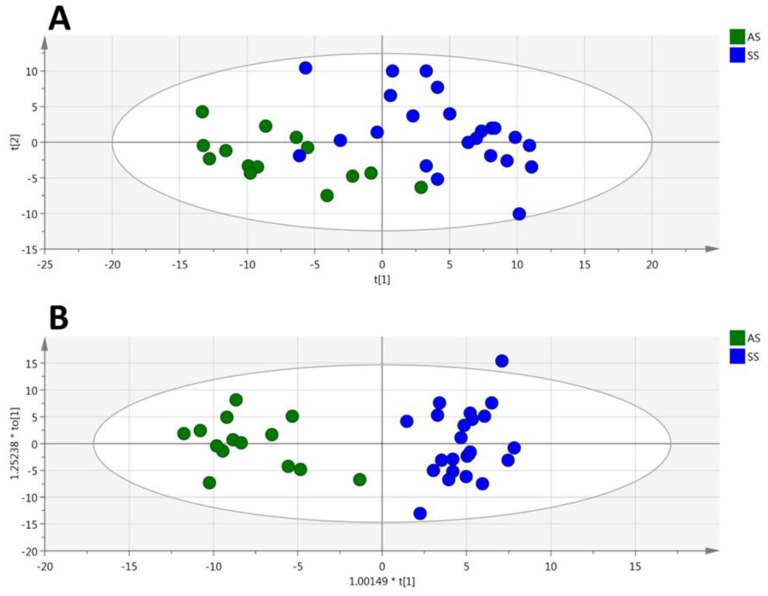
(RBC). PCA (**A**) and OPLS-DA (**B**) scatter plots obtained from the matrix of metabolites measured in the red blood cells from HbSS (blue circles, *n* = 23) and HbAS (green circles, *n* = 14) mice.

**Figure 4 cells-09-01334-f004:**
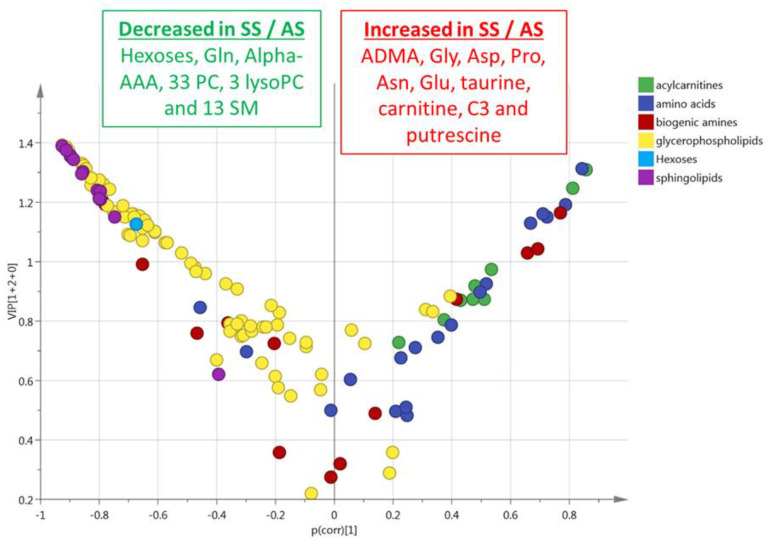
(RBC). Volcano plot (*p*_corr_ vs. VIP) from the OPLS-DA model shown in Figure 3B (red blood cells from HbSS vs. HbAS). Only the most discriminating metabolites with high VIP values ≥ 1 are listed in the boxes. Negative *p*_corr_ values (left) indicate diminished metabolite concentrations in HbSS versus HbAS genotypes, whereas positive *p*_corr_ values (right) indicate increased metabolite concentrations in HbSS versus HbAS genotypes.

**Figure 5 cells-09-01334-f005:**
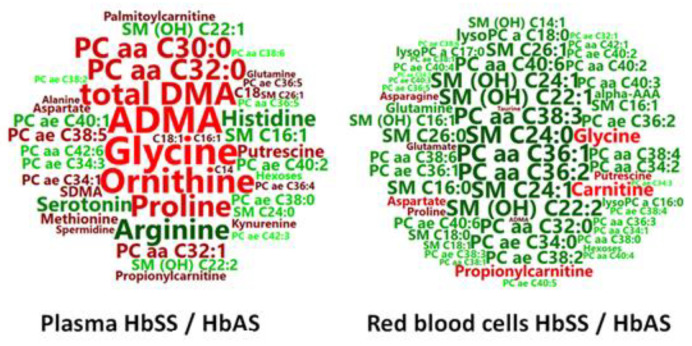
HbSS/HbAS signature summarization. Word clouds were calculated using metabolite VIPs to determine the size of the words, and loadings to determine the color scale, from dark green for the highest negative loadings (decreased in HbSS) and from deep red for the highest positive loadings (increased in HbSS).

**Figure 6 cells-09-01334-f006:**
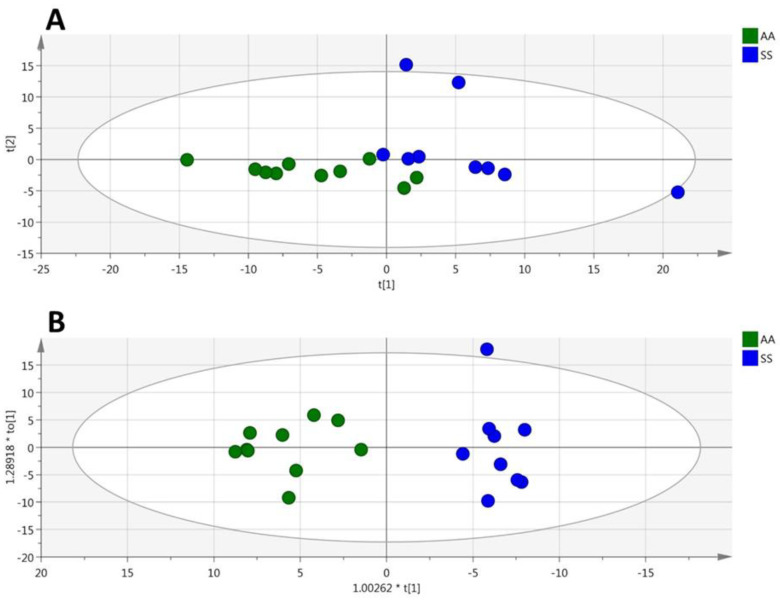
(plasma). PCA (**A**), OPLS-DA (**B**) scatter plots obtained from the matrix of metabolites measured in the plasmas from HbSS (blue circles, *n* = 9) and HbAA (green circles, *n* = 10) mice.

**Figure 7 cells-09-01334-f007:**
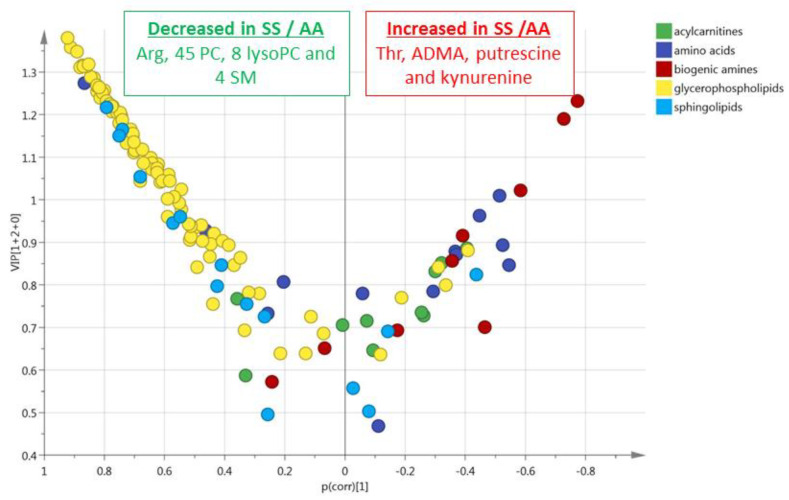
(plasma). Volcano scatter plots obtained from the matrix of metabolites measured in the plasmas from HbSS (blue circles, *n* = 9) and HbAA (green circles, *n* = 10) mice. Volcano plot (*p*_corr_ vs. VIP) from the OPLS-DA model shown in B (plasma from HbSS vs. HbAA). Only the most discriminating metabolites with high VIP values ≥ 1 are listed in the boxes. Negative *p*_corr_ values (left) indicate diminished metabolite concentrations in HbSS versus HbAA genotypes, whereas positive *p*_corr_ values (right) indicate increased metabolite concentrations in HbSS versus HbAA genotypes.

**Figure 8 cells-09-01334-f008:**
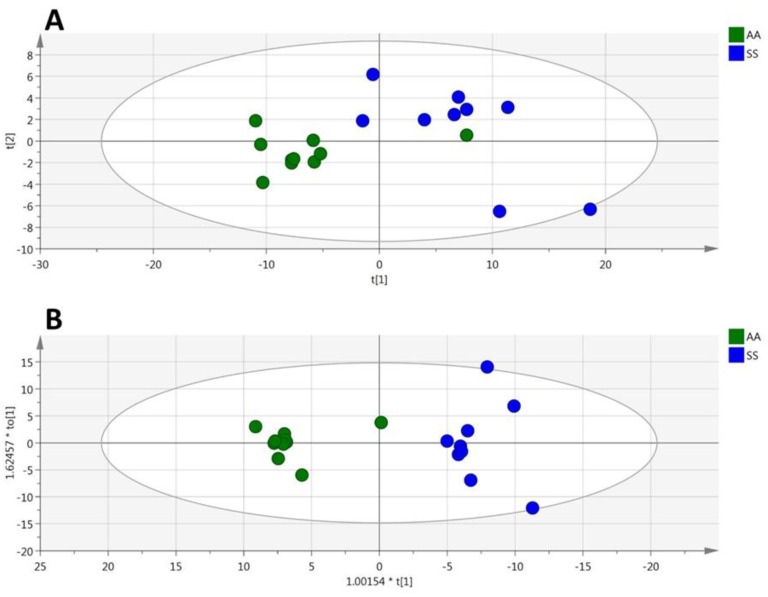
(RBC). PCA (**A**), OPLS-DA (**B**) scatter plots obtained from the matrix of metabolites measured in the red blood cells from HbSS (blue circles, *n* = 9) and HbAA (green circles, *n* = 10) mice.

**Figure 9 cells-09-01334-f009:**
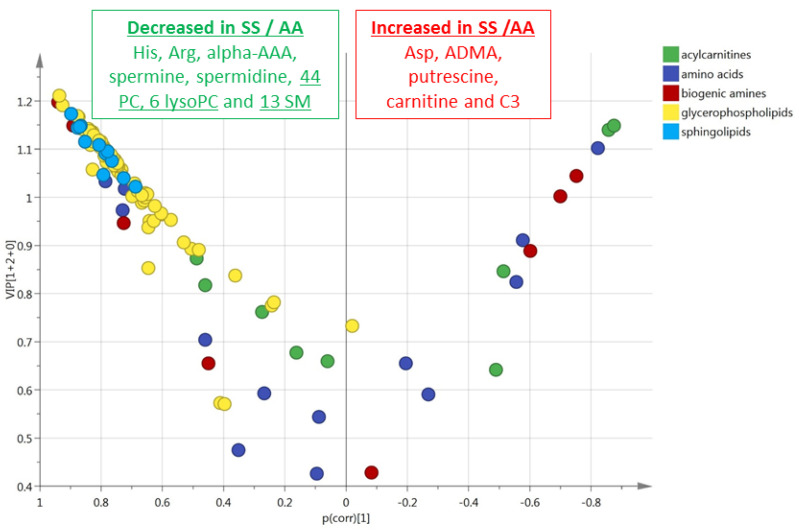
(RBC). Volcano scatter plots obtained from the matrix of metabolites measured in the red blood cells from HbSS (blue circles, *n* = 9) and HbAA (green circles, *n* = 10) mice. Volcano plot (*p*_corr_ vs. VIP) from the OPLS-DA model shown in B (red blood cells from HbSS vs. HbAA). Only the most discriminating metabolites with high VIP values ≥ 1 are listed in the boxes. Negative *p*_corr_ values (left) indicate diminished metabolite concentrations in HbSS versus HbAA genotypes, whereas positive *p*_corr_ values (right) indicate increased metabolite concentrations in HbSS versus HbAA genotypes.

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
