# Peer review of "Metabolomic Profiling of Plasma and Erythrocytes in Sickle Mice Points to Altered Nociceptive Pathways"

_cells, 2020, doi:10.3390/cells9061334_

Round 1

Reviewer 1 Report

The study is interesting and provides some background information for further study. It could be improved with additional summary/paragraph to correlate the difference in metabolites between diseased and health states to the sign and symptoms in human. The design of this animal model can be easily apply to human model.

Reviewer 2 Report

This article describes difference in the metabolic profile between mice carrying the homozygous human mutated haemoglobin sickle cell gene and control mice. The paper is well written, methods described in sufficient detail and findings discussed. The authors found differences in pathways related to nitric oxide synthesis/amino acids, bioenergetics, lipid remodelling and nociceptive pathways. Some of these differences have been found previously in other metabolic studies of sickle cell disease but this article builds on and adds further weight to these previous studies.

I only have a couple of minor queries and suggestions.

(1) A recent review of metabolomics in sickle cell disease is not referenced. Adebiyi et al 2019 Blood Adv (2019) 3 (8): 1347–1355. doi: 10.1182/bloodadvances.2018030619. This should be added.

(2) There are potentially significant gender differences between the HbSS and control groups (e.g. 8 males and 15 females in HbSS group vs 9 males and 5 females in the HbAS group). As gender differences have been described in sickle cell disease (such as Ceglie et al. Front. Mol. Biosci., 05 December 2019 | https://doi.org/10.3389/fmolb.2019.00140 ), could a comment be added regarding if there was any hypothesised reason for the gender bias of these groups and if there were any gender differences in metabolite profile, if these were studied.
